# Prevalence and Cost of Antipsychotic Prescribing, within the Context of Psycholeptic Prescribing, in the Irish Setting

**DOI:** 10.3390/healthcare12030338

**Published:** 2024-01-29

**Authors:** Muireann Vaughan, Siobhán Lucey, Laura J. Sahm

**Affiliations:** 1Pharmaceutical Care Research Group, School of Pharmacy, University College Cork, T12 K8AF Cork, Ireland; 2Department of Economics, Aras na Laoi, University College Cork, T12 T656 Cork, Ireland; s.lucey@ucc.ie; 3Pharmacy Department, Mercy University Hospital, T12 WE28 Cork, Ireland

**Keywords:** antipsychotic, psycholeptic, prescribing, cost analysis, drug schemes, time-trend analysis

## Abstract

Psycholeptic and specifically antipsychotic prescribing is increasing worldwide each year. This study aims to investigate the prevalence and cost of antipsychotic prescribing, within the wider frame of psycholeptic prescribing, in the Irish context. Quantitative analysis of a dataset from the Primary Care Reimbursement Service relating to cost and prescribing frequency of ATC Class N05 psycholeptic drugs from January 2020–August 2022 inclusive was conducted using Microsoft^®^ Excel^®^ for Microsoft 365 MSO (Version 2311) and STATA 18. Descriptive statistics and time-trend regression analysis were used to investigate the prescribing prevalence of psycholeptics and antipsychotics licensed for use in the Republic of Ireland, and the total cost per funding scheme. The prevalence of psycholeptic prescribing increased yearly from 2020–2022, peaking at 328,572 prescriptions in December 2020 with a total cost of psycholeptic drugs to the State in 2021 of €57,886,250, which was 0.5% of an increase on 2020. Over the 32-month time period, the average monthly cost of psycholeptic drugs was €4,436,469 on the General Medical Services (GMS) scheme and €369,154 on the Drug Payment Scheme (DPS). In 2021, quetiapine, olanzapine, and risperidone were the most prescribed antipsychotics, accounting for 66.58% of antipsychotics prescribed on the GMS scheme. This study identified the large expenditure on psycholeptics and antipsychotics in Ireland, with a higher proportion of the Irish healthcare budget spent on antipsychotics than that of the UK and the USA. The development of Irish antipsychotic prescribing guidelines may allow for structured, cost-effective prescribing.

## 1. Introduction

Worldwide trends show increased antipsychotic prescribing year-on-year. In the United Kingdom (UK), a study reported that antipsychotic prescribing rates doubled between 2007 and 2014 and that National Health Service (NHS) figures up to 2017 reinforced this trend [1]. Antipsychotics are primarily licensed for the treatment of psychosis disorders; however, they are increasingly used for the treatment of non-psychosis disorders including behavioural/psychological symptoms of dementia (BPSD), conduct disorder in children, and manic phases of bipolar disorder (BPD) [2,3,4]. In 2018, the lifetime risk of psychosis was estimated at 0.7% globally, yet the prevalence of antipsychotic use in the United States of America (USA) was 1.7% [5,6]. The growing off-label use of antipsychotics has been reported internationally as well as locally [6,7,8,9]. In 2019 the Health Service Executive (HSE) in Ireland launched a National Clinical Guideline on the appropriate prescribing of psychotropic medication for non-cognitive symptoms in dementia patients [9]. The guideline states that 1% of those with dementia, treated with an antipsychotic, will die due to the medication while only 20% of people experiencing non-cognitive symptoms will benefit from an antipsychotic [9,10].

Antipsychotics are broadly classified as first-generation antipsychotics (FGA), second-generation antipsychotics (SGA), and third-generation antipsychotics (TGA). They differ based on their side-effect profiles. FGA are more often associated with extrapyramidal side effects (EPSE), while SGA often lead to metabolic syndrome development requiring increased monitoring of metabolic parameters which can be burdensome on healthcare systems [11]. TGA are thought to have a more acceptable side-effect profile than FGA or SGA but are associated with akathisia [12]. The National Mental Health Services Survey (N-MHSS) 2020 in the USA highlighted the increased availability of SGA and TGA relative to FGA in the USA [13].

Psycholeptic drugs of the Anatomical Therapeutic Chemical (ATC) Classification system N05 are used to manage and treat a wide variety of illnesses and disorders including, schizophrenia, bipolar affective disorders, anxiety, and insomnia. 

There was a reported 25% increase in the prevalence of anxiety and depression worldwide during the Coronavirus disease 2019 (COVID-19) pandemic [14,15]. This likely contributed to an increase in z-hypnotics (N05C) prescribing in the USA, increased benzodiazepine prescribing (N05B) amongst a Catalan patient cohort, and the antipsychotic drug market growing by 10.8% globally during this timeframe [16,17,18]. First-episode psychosis (FEP) presentations increased in Ireland and the UK, as well as anxiety and depression presentations increasing worldwide in 2021 [14,19,20]. 

The cost associated with antipsychotic prescribing is significant. The global antipsychotic drugs market was valued at $14.54 billion (US dollars) in 2021 and was projected to grow to $15.50 billion in 2022 [18]. According to a research paper published by *The Lancet* in 2020, mental disorders as a whole are expected to cost the global economy $16 trillion by 2030 [21]. A 2019 study in the USA found that the total lifetime medical spending was $96,500 (24 percent) higher for a 25-year-old with a severe mental illness than for one without [22]. The extensive off-label use of antipsychotics is a large contributor to the increasing number and associated cost of antipsychotic prescriptions [23]. The cost of antipsychotics is largely funded by State schemes as the symptom burden associated with schizophrenia and psychosis disorders can preclude patients from retaining employment and affording medication [24,25]. Such schemes exist in most jurisdictions, with the NHS reimbursing medications in the UK and Medicaid available to low-income adults in the USA [26,27]. Studies conducted in the USA indicate that Medicaid reimbursement status affects antipsychotic prescribing prevalence [28,29,30].

In Ireland, whilst many reimbursement schemes exist, the two schemes most relevant to antipsychotic prescribing are the General Medical Services (GMS) scheme and the Drugs Payment Scheme (DPS). The GMS is a means-tested scheme based on income, expenses, marital status, and dependents. It should be noted that within the GMS scheme, age is also an important factor as those over 70 years have a different threshold for eligibility. The GMS scheme over-represents female, older, and more socially deprived populations [31]. Once eligibility is confirmed, patients are entitled to receive prescribed medicines free of charge other than a nominal prescription charge per item [32]. Under the DPS, an individual/family will pay no more than a specified limit each month; if the cost exceeds this amount, the state reimburses the pharmacy for the remainder of the cost [33]. The Primary Care Reimbursement Service (PCRS) is funded by the State and reimburses pharmacies for medications under both schemes. The PCRS had a budget of €3.269 bn in 2021. The percentage of the health budget spent on medication in Ireland in 2021 was 15.10%, in contrast to 10.27% of the UK budget and 8.79% on Medicaid in the USA [34,35,36,37,38,39]. The absence of Irish guidelines for the prescribing of antipsychotics for chronic conditions requiring antipsychotics, e.g., schizophrenia, may contribute to prescribing variations [40]. Currently, many international antipsychotic prescribing guidelines exist, including the National Institute for Health and Care Excellence (NICE) guidelines, UK, British Medical Journal (BMJ) Best Practice Guidelines, and Scottish Intercollegiate Guidelines Network (SIGN) guidelines [41,42,43]. Of note, there are no Irish guidelines in the context of a psychotic illness where chronic antipsychotic treatment is indicated. This study aims to investigate the prevalence and cost of antipsychotic prescribing, within the wider frame of psycholeptic prescribing, in the Irish setting.

## 2. Materials and Methods

This is an analysis of PCRS data provided to the authors after requesting data on ATC Class N05 prescribing from January 2020–August 2022. Data were aggregated to a monthly level over a time period. 

The PCRS provided this information with the following qualifiers: (i)The information provided is based on claim data which have been received by the PCRS from community pharmacists and include items reimbursed by PCRS only;(ii)The data do not capture items dispensed outside of community drug schemes where the prescription has been paid for privately by the patient or patient representative;(iii)These data may not capture claims which are under the DPS monthly threshold amount;(iv)PCRS does not capture data in relation to diagnosis or indication.

This was disaggregated by Anatomical Therapeutic Chemical Classification, Local Health Office (LHO), and Community Health Office (CHO) for each month of the time period January 2020 to August 2022; to conduct a monthly time-series analysis of the data at a national level, the data were aggregated for each month.

### 2.1. Data Collection

The PCRS provided access to data on the number of patients prescribed drugs within the ATC Class N05 psycholeptics including N05A—antipsychotics; N05B—anxiolytics; N05C—hypnotics and sedatives, with corresponding CHO numbers and LHO numbers [44]. 

### 2.2. Study Population

The data are pertaining to patients in the Republic of Ireland who were dispensed medications in ATC Class N05 who were registered on the GMS or DPS schemes between January 2020–August 2022. It should be noted that there is a risk of double-counting patients as a patient who received one drug may also have received a second drug within the same ATC classification. Dispensing fees, which are not standardised, are charges applied to medicines dispensed by pharmacies. Patients registered with the DPS contribute a designated amount towards their medication each month; if the cost exceeds this amount, the state reimburses the pharmacy for the remainder of the cost. The DPS thresholds are detailed in Appendix A. In 2021, 30.8% of the Irish population was registered on the GMS scheme and 1.47 million people were registered for the DPS, equivalent to 28.5% of the population [45].

Data relating to medications in ATC Class N05 paid for privately or relating to clozapine dispensing were not available. Data relating to the Long-Term Illness (LTI) scheme were available from the PCRS but were not requested for this analysis as, due to the conditions and related medications reimbursed under the LTI scheme, the number of antipsychotics reimbursed is likely to be minimal [46]. 

### 2.3. Data Analysis

Data were isolated from the raw data set and analysed using Microsoft^®^ Excel^®^ for Microsoft 365 MSO (Version 2311) with time-trend regression analyses conducted using STATA18.

#### 2.3.1. For the Period April 2020–April 2022 

The following subsets were compiled:A ranked list of the number of patients prescribed each drug in April 2020, 2021, and 2022 for both GMS and DPS schemes. This was the index month chosen as Ireland began lockdown in the middle of March 2020;A ranked list of the total ingredient cost per drug in April 2020 for both GMS and DPS schemes;A ranked list of the drug ingredient cost per patient in April 2020 (total ingredient cost of drug X/number of patients prescribed drug X) for both schemes [47];A list of the ingredient cost (cost of drug excluding dispensing fees and VAT on pharmacy claim) and total cost of each antipsychotic dispensed on GMS and DPS schemes in 2021.

The subsets of data were then analysed based on the following variables to determine potential factors with an influence on prescribing trends: Cost [ingredient cost and total cost];Period [April 2020 vs. April 2021 vs. April 2022].

#### 2.3.2. For the Period January 2020–August 2022

The following subsets were compiled:Monthly national totals for the number of prescription items and total cost per scheme (GMS and DPS);Monthly national totals for the number of prescription items and total cost (for both schemes combined);Annual national totals for the number of prescription items and total cost per scheme (GMS and DPS);Monthly national totals for the number of patients, number of prescriptions, and total cost per scheme (GMS and DPS) for the top four example drugs: paliperidone, quetiapine, olanzapine, and aripiprazole.Note: In the calculation of the monthly totals, an irregularity was noted in the HSE figures reported for May 2021 and June 2021; the source of this irregularity was due to the HSE cyber-attack in May 2021. Specifically, the figures reported for May were much lower than the previous months (due to the shutdown of all HSE systems in May) and the figures for June 2021 were much higher than the previous months (prior to May 2021) and subsequent months up to August 2022. To account for this anomaly, the figures for May and June 2021 were adjusted using an average of both months.

#### 2.3.3. For the Period January 2020–August 2022 

Descriptive statistics were obtained for each of these variables. Further, to show how each variable developed over this time period, a time-trend regression analysis represented as follows was employed: Y_t_ = β_0_ + β_1_Trend_t_ + ε_t_(1)
where the dependent variable Y_t_ is the respective monthly total for each of the above variables, the independent variable Trend_t_ is a time trend for the period January 2020 to August 2022, and ε_t_ is the error term for month t. Robust standard errors for Equation (1) were obtained using the Newey–West (1987) variance estimator which produces consistent standard errors for OLS regression coefficient estimates in the presence of autocorrelation in addition to possible heteroscedasticity [48]. The Newey–West variance estimator handles autocorrelation up to and including a lag of m; given the frequency of the data (monthly), results were obtained for m = 0, 1, 6, and 12.

## 3. Results

As per Table 1, results of the analysis of GMS data for April 2020 (a), April 2021 (b), and April 2022 (c) showed that the four most prescribed antipsychotics remained constant across the three years. Quetiapine, olanzapine, and risperidone were the most prescribed antipsychotics accounting for 66.58% of all antipsychotics prescribed, with quetiapine alone accounting for 35.19% of all patients prescribed antipsychotics.

The prevalence of SGA prescribing was greater than that of FGA prescribing, with a particular decrease in the number of haloperidol patients by 69.88% over the three-year period. There was an increase in the number of TGA patients, as highlighted by a 242.11% increase in the number of cariprazine patients over the period April 2020 to 2022.

In contrast, as displayed in Table 2, the most prescribed antipsychotics on the DPS in April 2020 (a), April 2021 (b), and April 2022 (c) were quetiapine, olanzapine, and lithium. While lithium patients decreased from April 2020 to April 2022 on the GMS scheme, they increased by 43.87% on the DPS. Risperidone was the sixth most prescribed antipsychotic, accounting for 6.15% of patients on the DPS in comparison to 10.43% of GMS patients. The GMS trend of increasing SGA/TGA patient numbers is similar in the DPS, with FGA accounting for only 15.52% of patients in April 2022.

As per Table 3, the results of the cost analysis of antipsychotics dispensed on GMS and DPS schemes in April 2020 showed paliperidone to be associated with the highest cost per patient on both schemes, with a total ingredient cost of €510,863.44 to the exchequer. The total ingredient cost of all antipsychotics dispensed in April 2020 was €1,922,115.56, with paliperidone accounting for 26.58% of this. TGA are among the more expensive agents, with cariprazine costing €132.10 per patient on the GMS scheme and aripiprazole also ranked within the top five most expensive antipsychotics. Lithium was the least expensive drug on the DPS costing €2.48 per patient. 

The total cost of antipsychotics to the State in 2021 was €34,010,700.50. Paliperidone was responsible for the highest spend on the GMS scheme and quetiapine was responsible for the same on the DPS. The total ingredient cost of all ATC Class N05A drugs was 69.03% of the GMS total cost and 71.49% of the DPS total cost. See Table 4.

Summary descriptive statistics were obtained for the GMS (Appendix A) and DPS (Appendix A) monthly totals, respectively, for all N05 drugs licensed in Ireland and the top four example N05 drugs over the 32-month observation period, including the average (Mean), a measure of variation (standard deviation—Std. Dev.), and the minimum (Min) and maximum (Max) values for the following variables: cost and number of prescription items (total and by example drug), and number of patients by example drug. It can be seen that the GMS averages are higher than the DPS in terms of cost, patients, and prescription items. 

The OLS results (with robust standard errors) of the time-trend regression analysis (Equation (1)) for all psycholeptic drugs (N05) in the GMS and DPS are reported in Table 5 and Table 6 respectively. Given that the findings of statistically significant/insignificant coefficient estimates were consistent across the various lag lengths (m = 0, 1, 6, and 12) for all regressions except for the variable “GMS Number of Prescription Items”, for brevity, the results for m = 0 are reported in Table 5 and Table 6 with the results for m = 1 reported for the variable “GMS Number of Prescription Items” (1 lag included to correct for 1st order negative autocorrelation). The GMS results in Table 5 show that whilst there was an insignificant change in total cost over the period of Jan 2020 to Aug 2022, there was a negative and statistically significant effect at the 5 percent level for the number of prescription items. 

When the top four example drugs in the GMS were analysed individually (results are not presented here but available on request) it was found that for paliperidone and quetiapine, the total cost, number of patients, and number of prescription items increased significantly for this time period whilst the numbers of patients and prescription items increased significantly for Aripiprazole. 

The corresponding results for the DPS are reported in Table 6 and in contrast to GMS, show that there was a positive and significant increase in both the total cost and number of prescription items for all psycholeptic drugs (N05) over the period of January 2020 to August 2022. Further, similar findings (results are not presented here but available on request) were obtained for each of the example drugs of paliperidone, quetiapine, olanzapine, and aripiprazole, with the time-trend results showing that they all significantly increased over the period of January 2020 to August 2022 on the DPS.

## 4. Discussion

In 2020, Ireland had a population of 4.985 million, increasing to 5.033 million in 2021 and 5.15 million in 2022. The prevalence of psycholeptic prescribing increased yearly from 2020–2022, peaking at 328,572 prescription items in December 2020. The minimum number of patients in Ireland receiving at least one antipsychotic on reimbursement schemes in April 2021 was 31,340. In 2021, 30.8% of the Irish population was registered on the GMS scheme [34]. Of these 1.54 million medical card holders, a minimum of 1.9% were prescribed antipsychotics. The total ingredient cost of antipsychotics on GMS and DPS schemes in 2021 was over €23.5 million. The total cost including dispensing fees and all associated costs was over €34 million. The PCRS budget for 2021 was €3.269 billion, therefore the 19 antipsychotics included account for approximately 1% of this budget [34,35]. The USA and UK also report that approximately 1% of their medication budget was spent on antipsychotics in 2021 [36,37,38,39]. However, the proportion of the Irish health budget allocated to medication (15.1%) is higher than that of the USA (10.27%) and the UK (8.79%). Therefore, Ireland had the highest proportional spend on antipsychotics in 2021 of all three jurisdictions [34,35,36,37,38,39]. The prescribing of antipsychotics during the time period of COVID-19 has been the subject of debate, with some research suggesting a strong association between exposure to antipsychotics and COVID-19 mortality [49]. A further study reported a spike in antipsychotic prescribing in the dementia and care home groups, which correlated with COVID-19 lockdowns and may have been due to prescribing of antipsychotics for palliative care. The same study, however, found that the monthly rate (per 1000 patients) associated with antipsychotic prescribing within the group of patients with serious mental illness, decreased slightly from 376.49 (95% CI 375.60 to 377.38) in Q1 2019 to 373.75 (95% CI 372.92 to 374.57) in Q4 2021 [50]. This is difficult to compare to our findings as diagnosis was not available as part of our dataset, but the overall spend on psycholeptics has increased since 2020.

Paliperidone was responsible for the highest spend on any antipsychotic across all three years, costing the exchequer €7.9 million through both schemes in 2021. The cost of paliperidone was €404.96 per patient on the GMS scheme in 2020. Two general features of antipsychotic drugs that are important to consider are (i) their potential for long-term use and (ii) the hesitance of prescribers to switch between antipsychotics once symptoms are controlled [51]. While TGA are among the more expensive agents, the body of evidence supporting their benefits in reducing metabolic side effects as well as their efficacy in negative symptom amelioration provides a clear justification for use [52]. From the literature it is clear which patient cohort(s) would benefit from TGA, allowing for an evidence-based rationale for this high-cost prescribing, in contrast to paliperidone, where such evidence appears to be lacking [53,54]. 

In this present study, quetiapine was found to be the most prescribed antipsychotic, irrespective of the funding scheme. This resulted in a cost of €7.35 million to the exchequer in 2021. The licensed indications for quetiapine in Ireland are the treatment of schizophrenia and BPD [55]. Quetiapine has a growing off-label use for the management of mood disorders as well as delirium and BPSD; these diagnoses may contribute to the high level of use seen across all three years [7,56]. The HSE have cautioned against quetiapine’s use in the elderly cohort due to its anticholinergic side-effect burden and associated stroke risk [2,9,57]. The SIGN and NICE guidelines also recommend quetiapine as a first-line agent for psychosis/hallucinations of Parkinson’s disease and Parkinson’s disease dementia [58,59]. The wide variety of licensed and off-label indications for quetiapine likely contribute to its continually growing popularity. Internationally, quetiapine prescribing rates are increasingly high. Quetiapine was the most prescribed antipsychotic in the USA in 2020, in Japan in 2014 and a UK study on antipsychotic prescribing from 2007–2014 also reported high and increasing levels of quetiapine use [60,61,62]. 

With the exception of clozapine, olanzapine is the antipsychotic associated with the highest burden of metabolic side effects and yet it was the second most prescribed antipsychotic on GMS and DPS schemes, costing €6.30 million to the State in 2021 [63,64]. Historically, olanzapine was viewed to be the most effective antipsychotic by prescribers and was also viewed favourably for its sedative properties [65]. However, a systematic review and meta-analysis of 402 randomised controlled trials in 2019 found that differences in efficacy between antipsychotics, in general, were small and uncertain [66]. Amisulpride, olanzapine, and risperidone were found to be equally effective, thus putting into question the narrative of olanzapine superiority [66]. Olanzapine use internationally is variable. In 2020 it was only the fourth most prescribed antipsychotic in the USA [61]. A study in 2014 mirrored this result, recording olanzapine as the fourth most prescribed antipsychotic in Japan and Hong Kong in those under 65 years old, citing a prevalence rate of between 0.5–1% per capita [60]. In contrast, in the UK in 2014 olanzapine was the most prescribed antipsychotic in primary care and in 2016 it was the second most prescribed antipsychotic in Scandinavia [67]. This indicates a variability in olanzapine prescribing between Europe and USA/Asia and a potential need for an increase in European TGA prescribing in line with other jurisdictions to reduce antipsychotic-induced metabolic syndrome.

Risperidone was the third most prescribed antipsychotic across the study period. It has a slightly broader range of indications than other SGA, including a licensed indication for the short-term treatment (up to 6 weeks) of persistent aggression in patients with moderate/severe Alzheimer’s dementia [68]. In reality, this duration of treatment is often extended [9]. It is also one of the few antipsychotics licenced for use in children from the age of five years with sub-average intellectual functioning for the short-term treatment of persistent aggression [68]. Globally, risperidone was the third most prescribed antipsychotic in the USA in 2020 [61]. Interestingly, risperidone use decreased by 28% in Australia after licensing changes in 2015 amended the maximum duration of use in dementia patients from an indefinite duration to 12 weeks, followed by guideline updates reflecting this change in 2016 and 2019 [8,69,70]. 

Both aripiprazole and cariprazine grew in popularity year-on-year from 2020 to 2022, with more than three times the number of cariprazine prescriptions dispensed on the GMS in April 2022 than in April 2020 [52]. Internationally, aripiprazole use is increasing. Aripiprazole was the second most prescribed antipsychotic in the USA in 2020 and it was one of the five most prescribed antipsychotics as monotherapy in a Spanish study in 2015 [61,71]. This popularity may be attributed to guidelines proposing aripiprazole for first-line treatment of FEP [53,72]. Another reason for the increase in TGA prescriptions may be related to clinical trials that have shown cariprazine to be superior to all other antipsychotics including risperidone and amisulpride in ameliorating negative symptoms [73]. Historically, negative symptoms of schizophrenia have been challenging to target, with most FGA and SGA failing to achieve a meaningful reduction in negative symptoms on the positive and negative syndrome scale in clinical trials [74,75]. 

FGA prescribing decreased on GMS and DPS schemes across the study period as highlighted by an almost 70% decrease in haloperidol prescriptions across the timeframe. The EPSE associated with FGA are prominent and bothersome side effects. Therefore, the reduction in FGA use could be explained by increased patient awareness of such and consequent refusal to accept treatment with these adverse effects [76]. Ireland is not alone in its trend towards the preferential use of SGA; FGA use is decreasing globally, which may lead to a reduced level of clinical experience with FGA [12,67,77]. 

Finally, current practice in Ireland may benefit from having a set of national guidelines to address the appropriate use of antipsychotic medications for those who require them for a chronic psychotic disorder. Until now, no specific guidelines for antipsychotics have been developed in Ireland and presently, clinical practice is sometimes based on international guidelines, but only where the healthcare professional is aware of these. Patients who have a psychotic illness are a cohort that is likely to have a medication regimen that needs to be tailored to patients’ individual needs. The development of Irish antipsychotic prescribing guidelines will allow prescribers to access the evidence base to ensure that these needs are met. This will help to address the prescribing variances identified in this study. 

### Strengths and Limitations

This is the first study that the authors are aware of that has examined the prescribing and cost of psycholeptic and specifically antipsychotic medications in the Irish context over an extended period of 32 months. Further, the study provides a time-series analysis of this 32-month period which incorporates a pandemic and the effect of the same on these drugs in terms of cost, and prescription numbers nationally and for the top four N05 drugs. The study is not without its limitations, however, and one of the main drawbacks to the currently available data is the lack of a diagnosis/indication for the medication being prescribed. This makes the interpretation of the data as a function of diagnosis impossible to undertake. In addition, whilst it is best practice to prescribe monotherapy where clinically appropriate, it is possible, and in fact likely, that some patients were receiving more than one medication in the classes observed. Finally, as demographic data including age and gender were not requested, it was not possible to investigate any correlations due to these variables.

## 5. Conclusions

This study has established that psycholeptics and specifically antipsychotics impose a high cost on the Irish exchequer. The amount of the Irish health budget that was spent on antipsychotics in 2021 was proportionately higher than that of the USA and the UK. The rate of antipsychotic prescribing, specifically SGA prescribing, increased each year from April 2020–2022. The appropriateness of this increased rate of prescribing and its associated cost are hard to estimate as the indications for medication are not centrally recorded in Ireland. There is also a lack of Irish guidelines in this area to inform prescribing practices.

In summary, it would be advantageous to have a set of national guidelines on the appropriate use of antipsychotic medications for those who require them for a chronic psychotic disorder. As mentioned, guidelines will also increase the audit potential of prescribing practices nationally.

A further recommendation would be to introduce shared information technology systems, recording the indications of medication, for those prescribing and those dispensing medications. This is current practice in the UK and Norway and allows for pharmacist clinical interventions as well as nationwide audits of antipsychotic prescribing appropriateness. These interventions have the potential to improve medication safety, cost-effective prescribing and the quality of care provided to patients, thus resulting in better outcomes for all.

## Figures and Tables

**Table 1 healthcare-12-00338-t001:** Ranked list of ATC Class N05A drugs dispensed on the GMS scheme by total number of patients in (a) April 2020, (b) April 2021, (c) April 2022.

April 2020 (a)			April 2021 (b)			April 2022 (c)		
Drug	Medicine Type	Patient Number	Drug	Medicine Type	Patient Number	Drug	Medicine Type	Patient Number
Quetiapine	SGA	25,876	Quetiapine	SGA	28,807	Quetiapine	SGA	29,487
Olanzapine	SGA	16,714	Olanzapine	SGA	17,622	Olanzapine	SGA	17,096
Risperidone	SGA	8371	Risperidone	SGA	8390	Risperidone	SGA	8142
Prochlorperazine	FGA	7490	Prochlorperazine	FGA	7954	Prochlorperazine	FGA	7890
Lithium	MS	5341	Aripiprazole	TGA	5701	Aripiprazole	SGA	5785
Aripiprazole	TGA	5234	Lithium	MS	5385	Lithium	MS	5084
Chlorpromazine	FGA	1920	Chlorpromazine	FGA	1824	Chlorpromazine	FGA	1682
Amisulpride	SGA	1382	Amisulpride	SGA	1431	Amisulpride	SGA	1356
Haloperidol	FGA	1318	Paliperidone	SGA	1241	Paliperidone	SGA	1291
Paliperidone	SGA	1192	Zuclopenthixol	FGA	709	Zuclopenthixol	FGA	664
Zuclopenthixol	FGA	674	Haloperidol	FGA	653	Flupentixol	FGA	548
Flupentixol	FGA	586	Flupentixol	FGA	603	Haloperidol	FGA	397
Sulpiride	FGA	356	Sulpiride	FGA	350	Sulpiride	FGA	330
Asenapine	SGA	90	Asenapine	SGA	81	Asenapine	SGA	87
Ziprasidone	SGA	77	Ziprasidone	SGA	78	Ziprasidone	SGA	82
Perphenazine	FGA	21	Cariprazine	TGA	39	Cariprazine	TGA	65
Cariprazine	TGA	19	Perphenazine	FGA	21	Perphenazine	FGA	22
Promazine	FGA	17	Promazine	FGA	14	Promazine	FGA	14
Trifluoperazine	FGA	16	Trifluoperazine	FGA	0	Trifluoperazine	FGA	0

Key: GMS—General Medical Services, FGA—First Generation Antipsychotic, SGA—Second Generation Antipsychotic, TGA—Third Generation Antipsychotic, MS—Mood Stabiliser.

**Table 2 healthcare-12-00338-t002:** Ranked list of ATC Class N05A drugs dispensed on the DPS by total number of patients in (a) April 2020, (b) April 2021, (c) April 2022.

April 2020 (a)			April 2021 (b)			April 2022 (c)		
Drug	Medicine Type	Patient Number	Drug	Medicine Type	Patient Number	Drug	Medicine Type	Patient Number
Quetiapine	SGA	2144	Quetiapine	SGA	2533	Quetiapine	SGA	3523
Olanzapine	SGA	1229	Olanzapine	SGA	1410	Olanzapine	SGA	1957
Lithium	MS	978	Lithium	MS	1039	Lithium	MS	1407
Prochlorperazine	FGA	819	Prochlorperazine	FGA	979	Prochlorperazine	FGA	1340
Aripiprazole	TGA	446	Aripiprazole	TGA	552	Aripiprazole	TGA	739
Risperidone	SGA	414	Risperidone	SGA	474	Risperidone	SGA	576
Amisulpride	SGA	86	Amisulpride	SGA	102	Amisulpride	SGA	121
Paliperidone	SGA	73	Paliperidone	SGA	79	Paliperidone	SGA	90
Sulpiride	FGA	55	Sulpiride	FGA	53	Sulpiride	FGA	68
Haloperidol	FGA	49	Chlorpromazine	FGA	49	Chlorpromazine	FGA	59
Chlorpromazine	FGA	46	Haloperidol	FGA	29	Haloperidol	FGA	31
Asenapine	SGA	14	Ziprasidone	SGA	18	Asenapine	SGA	21
Zuclopenthixol	FGA	12	Flupentixol	FGA	17	Trifluoperazine	FGA	19
Flupentixol	FGA	11	Asenapine	SGA	16	Ziprasidone	SGA	18
Ziprasidone	SGA	10	Zuclopenthixol	FGA	15	Flupentixol	FGA	18
Trifluoperazine	FGA	8	Cariprazine	TGA	8	Zuclopenthixol	FGA	14
Cariprazine	TGA	5	Trifluoperazine	FGA	6	Cariprazine	TGA	8
Perphenazine	FGA	*	Perphenazine	FGA	*	Perphenazine	FGA	*
Promazine	FGA	*	Promazine	FGA	*	Promazine	FGA	*

Key: DPS—Drugs Payment Scheme, FGA—First Generation Antipsychotic, SGA—Second Generation Antipsychotic, TGA—Third Generation Antipsychotic, MS—Mood Stabiliser. * indicates fewer than five patients.

**Table 3 healthcare-12-00338-t003:** Ranked list of ATC Class N05A drugs dispensed on GMS and DPS schemes by (i) ingredient cost per patient and (ii) total ingredient cost in April 2020.

	April 2020 GMS					April 2020 DPS					
Rank	Drug	Ingredient Cost Per Patient *	Rank	Drug	Total Ingredient Cost	Rank	Drug	Cost Per Patient to the State **	Rank	Drug	Total Ingredient Cost *
1	Paliperidone	€404.96	1	Paliperidone	€482,710.78	1	Paliperidone	€385.65	1	Quetiapine	€28,329.93
2	Cariprazine	€132.10	2	Olanzapine	€329,215.33	2	Asenapine	€110.79	2	Paliperidone	€28,152.66
3	Asenapine	€102.34	3	Quetiapine	€305,715.35	3	Cariprazine	€107.35	3	Aripiprazole	€26,138.75
4	Ziprasidone	€102.13	4	Aripiprazole	€298,571.72	4	Ziprasidone	€85.59	4	Olanzapine	€19,810.42
5	Aripiprazole	€57.04	5	Risperidone	€207,571.52	5	Aripiprazole	€58.61	5	Risperidone	€12,868.26
6	Promazine	€46.08	6	Amisulpride	€61,221.40	6	Amisulpride	€33.86	6	Amisulpride	€2911.97
7	Amisulpride	€44.30	7	Prochlorperazine	€19,012.18	7	Risperidone	€31.08	7	Lithium	€2421.48
8	Risperidone	€24.80	8	Haloperidol	€18,542.26	8	Trifluoperazine	€23.24	8	Prochlorperazine	€2135.92
9	Sulpiride	€20.67	9	Lithium	€15,225.72	9	Olanzapine	€16.12	9	Asenapine	€1551.07
10	Perphenazine	€20.65	10	Chlorpromazine	€14,922.29	10	Sulpiride	€15.71	10	Sulpiride	€863.98
11	Olanzapine	€19.70	11	Asenapine	€9210.30	11	Quetiapine	€13.21	11	Ziprasidone	€855.90
12	Haloperidol	€14.07	12	Ziprasidone	€7864.11	12	Haloperidol	€12.22	12	Haloperidol	€598.77
13	Quetiapine	€11.81	13	Sulpiride	€7357.96	13	Flupentixol	€9.63	13	Cariprazine	€536.76
14	Zuclopenthixol	€10.73	14	Zuclopenthixol	€7230.41	14	Perphenazine	€9.15	14	Chlorpromazine	€253.97
15	Flupentixol	€10.37	15	Flupentixol	€6077.37	15	Zuclopenthixol	€8.39	15	Trifluoperazine	€185.89
16	Chlorpromazine	€7.77	16	Cariprazine	€2509.92	16	Promazine	€7.82	16	Flupentixol	€105.96
17	Trifluoperazine	€6.29	17	Promazine	€783.30	17	Chlorpromazine	€5.52	17	Zuclopenthixol	€100.63
18	Lithium	€2.85	18	Perphenazine	€433.64	18	Prochlorperazine	€2.61	18	Perphenazine	€9.15
19	Prochlorperazine	€2.54	19	Trifluoperazine	€100.71	19	Lithium	€2.48	19	Promazine	€7.82
				Total	€1,794,276.27					Total	€127,839.29

Key: * Ingredient cost: cost of drug excluding dispensing fees and VAT on pharmacy claim. ** In April 2020, patients on the DPS scheme paid the first €124 per month for their medication. The differences in the ingredient cost per patient between GMS and DPS schemes are due to patients contributing a higher amount towards the cost of their medication on the DPS.

**Table 4 healthcare-12-00338-t004:** Class N05A drugs dispensed on GMS and DPS schemes ranked in order of total cost with corresponding ingredient cost for 2021.

GMS 2021			DPS 2021		
Drug	Ingredient Cost * (€)	Total Cost (€)	Drug	Ingredient Cost ** (€)	Total Cost (€)
Paliperidone	€6,089,237.81	€7,434,405.78	Quetiapine	€370,750.97	€568,540.15
Quetiapine	€3,834,030.50	€6,776,960.12	Paliperidone	€401,287.78	€487,350.40
Olanzapine	€3,996,869.30	€5,915,189.54	Aripiprazole	€367,191.67	€435,921.17
Aripiprazole	€3,724,845.48	€4,513,776.27	Olanzapine	€269,432.18	€380,397.68
Risperidone	€2,224,589.90	€3,277,722.22	Risperidone	€131,968.88	€185,902.62
Amisulpride	€731,814.64	€892,447.65	Lithium	€30,070.70	€117,802.20
Lithium	€173,895.30	€776,559.56	Prochlorperazine	€28,379.85	€84,985.52
Prochlorperazine	€231,891.23	€718,631.84	Amisulpride	€41,918.87	€49,288.87
Chlorpromazine	€162,136.54	€382,728.73	Ziprasidone	€19,076.27	€20,685.27
Haloperidol	€128,276.79	€207,516.67	Asenapine	€17,444.28	€18,638.28
Zuclopenthixol	€87,139.45	€159,832.75	Cariprazine	€14,655.06	€15,418.06
Flupentixol	€71,203.31	€132,542.93	Haloperidol	€10,209.58	€13,701.81
Sulpiride	€80,296.93	€111,682.19	Sulpiride	€10,046.69	€13,362.19
Asenapine	€102,619.44	€109,246.77	Chlorpromazine	€2777.87	€6267.37
Ziprasidone	€95,378.61	€104,376.02	Trifluoperazine	€3036.74	€3605.24
Cariprazine	€72,311.40	€77,452.02	Flupentixol	€1813.45	€3394.04
Perphenazine	€4435.23	€6260.01	Zuclopenthixol	€1180.74	€2244.95
Promazine	€4255.72	€5022.72	Promazine	€338.46	€433.46
Trifluoperazine	€7.57	€24.84	Perphenazine	€254.59	€382.59
Total	€21,815,235.15	€31,602,378.63	Total	€1,721,834.63	€2,408,321.87

Key: * Ingredient cost: cost of drug excluding dispensing fees and VAT on pharmacy claim. ** In 2021, patients on the DPS scheme paid the first €114 per month for their medication.

**Table 5 healthcare-12-00338-t005:** All N05 drugs in the GMS scheme from January 2020–August 2022.

Variables	(1)	(2)
	GMS Total Cost	GMS No. of Prescription Items
Trend	−3353.866	−248.316 **
	(2247.817)	(102.904)
	[−7944.520–1236.788]	[−458.473–−38.159]
Observations	32	32

Robust standard errors in parentheses (robust standard errors were obtained using the Newey–West (1987) variance estimator); 95% confidence intervals in brackets. ** *p* < 0.05.

**Table 6 healthcare-12-00338-t006:** All N05 drugs in the DPS scheme from January 2020–August 2022.

Variables	(1)	(2)
	DPS Total Cost	DPS No. of Prescription Items
Trend	3994.006 ***	388.621 ***
	(475.326)	(50.886)
	[3023.261–4964.751]	[284.698–492.544]
Observations	32	32

Robust standard errors in parentheses (robust standard errors were obtained using the Newey–West (1987) variance estimator); 95% confidence intervals in brackets. *** *p* < 0.01.

## Data Availability

Data were requested for research purposes for the analysis within this study and permission was neither sought nor granted from the PCRS for the data to be made publicly available.

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
