# Peer review of "Prevalence and Cost of Antipsychotic Prescribing, within the Context of Psycholeptic Prescribing, in the Irish Setting"

_healthcare, 2024, doi:10.3390/healthcare12030338_

Round 1

Reviewer 1 Report

Comments and Suggestions for Authors

The paper considers the use of antipsychotics in Ireland using pharmacy claims data. There are some interesting insights in this mainly descriptive paper.

 There are some comments to consider.

Introduction: The authors state ‘In Ireland, two reimbursement schemes exist, the General Medical Services (GMS) scheme and the Drugs Payment Scheme (DPS). The GMS is a means-tested scheme based on income, expenses, marital status, and dependents’ There are more than two reimbursement schemes in Ireland (LTI, High tech). I think the authors are referring to those relevant to anti-psychotics. Also age is important in the GMS scheme as those over 70 years have a different threshold for eligibility. The GMS over-represents female, older and more socially deprived populations.

The authors state ‘The absence of clinical guidelines is a key contributor to prescribing variations.’ Is this specifically for anti-psychotics ? – clinical guidelines do exist for other conditions/treatments. Also are prescribers not using other guidelines e.g. NICE? https://www.hse.ie/eng/about/personalpq/pq/2023-pq-responses/january-2023/pq-4146-23-mark-ward.pdf guideline for use in dementia

https://www.hse.ie/eng/dementia-pathways/files/national-clinical-guideline-no-21-appropriate-prescribing-of-psychotropic-medication-for-non-cognitive-symptoms-in-people-with-dementia.pdf]

Methods

The authors should use the STROBE checklist for reporting and provide the study design (repeated cross-sectional study?)

The number of patients is used as a metric however, this does not account for overlap of patients when on more than one type of drug. This needs to be explained further, e.g. adding the numbers of patients is potentially double-counting the same patients.

Data relating to medications in ATC Class N05 reimbursed under the Long-Term Illness scheme, paid for privately or relating to clozapine dispensing were not available. – LTI data are available if requested. However, the numbers on AP reported in the LTI scheme will likely be low.

No mention of the lack of indication available so it is not known for what indication prescribed these drugs. Also no mention of ages included, were differences across age considered? Would it make more sense to exclude children? And no mention of gender differences.

Descriptive statistics were obtained for each of these variables. – what descriptive stats are used and which variables?

Rates of use per population (/1000 pop) would be better than crude numbers as the underlying population (i.e. GMS) may vary over time.

Time trend analysis – was there any adjustment for autocorrelation in the data? The same people tend to be prescribed month to month?

Not clear why two different periods considered ‘For the period April 2020- April 2022’ and ‘Jan 2020-Aug 2022’.  It would have been interesting to consider pre-covid and post-covid comparisons e.g. going back to include 2018 and 2019 data.

Page 3 line 124 ‘list of the ingredient cost and total cost of each antipsychotic’ – should explain the difference in costs here.

Why is April 2020 (and April 2021 etc) chosen for drug costs? May not be representative as a month after the covid-19 lockdown?

Results

Would be helpful for international audience to include the size of the underlying population and rates so that comparable to other settings/countries?

Table 2 (and others) – should not present any data where the  numbers <5 to avoid possibility of identification. These should be removed and replaced with a symbol or footnote.

Table 3. Ranked list of ATC Class-N05A drugs dispensed on GMS and DPS schemes

But the heading in the table say GMS – confusing – what is included in  the table ?

Also, the different in ingredient cost/patient and cost to state is in the dispensing fees and VAT -but the costs seem higher for the ingredient cost- is this correct? The use of dispensing fees and thresholds for payments under the DPS would be better described earlier in the methods. Also the DPS threshold changes so this could be described in methods

Page 7 – The authors state ‘as expected, the GMS averages are higher than the DPS ‘ why is it as expected?

The tables are quite detailed and lots of them – could some of these be in an appendix ?

Costs tend to be skewed - Means/SD may not be  appropriate.

Is it prescriptions or items on prescriptions? – it is possible to have more than one item on a prescription – e.g. a drug may have different doses. The PCRS provides items (unless the number of claims is used)

Table 7 and table 8. All NO5 Drugs = typo N05

Table 7 – why is p<0.10 used for significance? This is not standard and over-inflates the type 1 error, it should not be used. The constant term is a nuisance variable and should not be presented (unless used for prediction).  95% Confidence intervals would be helpful.

Robust standard errors are referred to but not explained why they are used in the methods

Discussion

‘Finally, current practice in Ireland may over-rely on psycholeptic medications’ – why?

No strengths and limitations of the research provided. Also implications missing from the discussion, mentioned in the conclusion.

No mention of the impact of Covoid-19 on prescribing in the discussion which is the period under study. There were studies linking AP to covid mortality at the time (https://www.thelancet.com/journals/lanpsy/article/PIIS2215-0366(21)00396-5/fulltext

And others observed spikes in AP use during pandemic mainly associated with dementia and care homes: https://mentalhealth.bmj.com/content/26/1/e300775

Author Response

Thank you for your extensive and valuable feedback and review of your manuscript.

Please find attached our response to each of your suggestions.

Yours faithfully

Laura Sahm

Reviewer 2 Report

Comments and Suggestions for Authors

Major Concern:

1. The paper lacks a comprehensive description of the dataset used in the study. A detailed account of the dataset, including  its origin, size, characteristics, and any preprocessing steps, is crucial for the readers to understand the basis of the study. This information is essential for reproducibility and to assess the generalizability of the findings.

2. The discussion section inadequately addresses the limitations of the study. It is essential to explicitly acknowledge and discuss any potential constraints, biases, or challenges associated with the research design and methodology. This inclusion will provide a more balanced interpretation of the findings and enhance the overall credibility of the study.

Minor Concern:

1. The paper lacks statements regarding ethical approval for the study. It is imperative to explicitly state whether ethical guidelines were followed and provide details on the approval process, especially when involving human subjects or sensitive data. This information is crucial for establishing the ethical soundness of the research and ensuring compliance with ethical standards.

Author Response

Thank you for your time ad effort in the review of our manuscript.

Please find attached our point by point response.

Yours faithfully

Laura Sahm

Round 2

Reviewer 1 Report

Comments and Suggestions for Authors

The majority of the comments have been addressed. There are only one or two minor corrections suggested to the text.

1. The footnote 2 in Table 5 is very long and would be better placed in the methods section and replacing the long footnote 2 with a shorter note.

2. In the revised discussion the limitations include '..therefore a total number of patients receiving psycholeptics or antipsychotics cannot be reliably estimated.' This is not strictly true. It is possible to obtain this information from the PCRS (they can aggregate number of patients at different levels of ATC), so I suggest this statement is removed. Also in the limitations ' Finally, the absence of demographic data including age and gender render it impossible to investigate any correlations due to these variables"  Information on age and gender is available via the HSE-PCRS - the authors had not specifically requested this but it is possible to obtain.

In the discussion section relating to coviid-19 impact now added, the authors state '...found that the monthly rate (per 1000 patients) of antipsychotic prescribing within..'  Perhaps change this to '...found that the monthly rate (per 1000 patients) associated with antipsychotic prescribing within...'

Author Response

Thank you so much for your time in a further review of our manuscript.

We have amended as attached.

Yours faithfully

Laura Sahm
